# Effect of spironolactone on survival in patients undergoing maintenance hemodialysis

**Seok Hui Kang[1], Bo Yeon Kim[2], Eun Jung Son[3], Gui Ok Kim[3], Jun Young Do[1]\***

**1** Department of Internal Medicine, Division of Nephrology, College of Medicine, Yeungnam University, Daegu, Republic of Korea, **2** Healthcare Review and Assessment Committee, Health Insurance Review and Assessment Service, Wonju, Republic of Korea, **3** Quality Assessment Department, Health Insurance Review and Assessment Service, Wonju, Republic of Korea

\* jydo@med.yu.ac.kr

## Abstract

### Background

Previous studies have reported inconsistent results regarding the advantages or disadvantages of spironolactone use in patients undergoing hemodialysis (HD). This study aimed to evaluate survival according to the use of spironolactone in a large sample of patients undergoing maintenance HD.

### Methods

This retrospective study used laboratory and clinical data from the national HD Quality Assessment Program and claims data. The participants of the quality assessment program were patients who had been undergoing maintenance HD for $\geq$ 3 months, patients undergoing HD at least twice a week. Patients with no spironolactone prescription during the assessment periods were designated as the control group. Patients with one or more prescriptions of spironolactone during the assessment periods were assigned to the SPR group.

### Results

The number of patients in the control and SPR groups were 54,588 and 315, respectively. The 5-year survival rates were 69.1% and 59.1% in the control and SPR groups, respectively ($P < 0.001$). Cox regression analyses showed that the hazard ratio in the SPR group was 1.34 ($P < 0.001$) in univariate analysis and 1.13 ($P = 0.249$) in multivariable analysis. Univariate Cox-regression analysis showed a better patient survival rate in the control group than in the SPR group; however, multivariable analyses showed similar patient survival rates between the two groups.

### Conclusion

This study showed no difference in survival between patients undergoing HD with and without spironolactone use.

**Data Availability Statement:** The raw data were generated at the Health Insurance Review and Assessment Service. The database can be requested from the Health Insurance Review and Assessment Service by sending a study proposal

including the purpose of the study, study design, and duration of analysis through an e-mail (turtle52@hira.or.kr) or at the web site (https://www.hira.or.kr). The authors cannot distribute the data without permission.

**Funding:** This work was supported by the Medical Research Center Program through the National Research Foundation (NRF) of Korea funded by the Ministry of Science, ICT, and Future Planning (2022R1A5A2018865, SHK), the Basic Science Research Program through the NRF of Korea, funded by the Ministry of Education (2022R1I1A3072966, SHK), and the NRF grant funded by the Korea government (MSIT) (2022R1F1A1076151, JYD). The funders had no role in the design of the study; in the collection, analyses, or interpretation of data; in the writing of the manuscript; or in the decision to publish the results.

**Competing interests:** The authors have declared that no competing interests exist.

## Introduction

Kidney damage can occur as a result of various factors, including high glucose levels, ischemia, or use of medications. Chronic injury to the kidneys can lead to the development of chronic kidney disease which can progress to end-stage renal disease requiring renal replacement therapy. Renal replacement therapy includes hemodialysis (HD), peritoneal dialysis, and kidney transplantation, with HD being the most commonly used modality. Patients undergoing HD have higher mortality rates than the general population of patients without dialysis [1]. The leading cause of death in patients undergoing HD is cardiovascular disease. The proportion of patients who died from cardiovascular disease in patients undergoing HD is approximately 51.5% in the United States and 43.6% in South Korea [2,3]. Therefore, many studies have focused on the diagnosis or treatment of cardiovascular disease in patients undergoing HD. However, the evidence is insufficient on whether treatments for cardiovascular disease with a survival benefit in patients without dialysis similarly benefit patients undergoing HD.

Spironolactone is a well-known mineralocorticoid receptor antagonist with various beneficial effects, such as attenuation of myocardial fibrosis, left ventricular hypertrophy, or arrhythmia, which in turn is associated with a survival benefit in patients without dialysis [4]. However, for patients undergoing HD with no residual renal function, the use of spironolactone has been associated with adverse effects, including hypotension and hyperkalemia [4]. Previous studies have reported inconsistent results regarding the advantages or disadvantages of spironolactone use in patients undergoing HD. Additional studies on the effects of spironolactone on patient survival are needed to definitely establish the risk or benefits of spironolactone use in patients undergoing HD. This study aimed to evaluate survival according to the use of spironolactone in a large sample of patients undergoing maintenance HD.

## Patients and methods

### Data source and study population

This retrospective study used laboratory and clinical data from the national HD Quality Assessment Program and claims data from the Health Insurance Review and Assessment (HIRA) of South Korea [5,6]. Briefly, the fourth, fifth, and sixth iterations of the HD Quality Assessment Program were conducted in July-December 2013, July-December 2015, and March-August 2018, respectively. The participants of the quality assessment program were patients who had been undergoing maintenance HD for ≥ 3 months, patients undergoing HD at least twice a week (eight or more sessions per month), and patients aged ≥18 years. We analyzed the HD quality assessment data and claims data of all patients who participated in HD quality assessments by HIRA. The data of the fourth, fifth, and sixth HD Quality Assessment program were collected at May-Jun 2014, May-July 2016, and January-February 2019, respectively. The data for claims or death were collected at April-May 2022. These data were assessed between May 2022 and April 2023 for research purpose.

The numbers of patients who participated in the fourth, fifth, and sixth iterations of the HD Quality Assessment Program were 21,846, 35,538, and 31,294, respectively. Among them, we excluded repeat participants (n = 32,440), those with insufficient data, and those who were undergoing HD using a catheter (n = 1,335). Finally, 54,903 patients were included in the analysis. This study was approved by the Institutional Review Board of Yeungnam University Medical Center (approval no. YUMC 2022-01-010), which waived the requirement for informed consent because of the retrospective study design and the anonymization and de-identification of patient records and information before the analysis.

## Study variables

Clinical data, including age, sex, underlying cause of end-stage renal disease, and type of vascular access, were collected. Laboratory data during the assessments, including hemoglobin (g/dL), $Kt/V_{urea}$, serum albumin (g/dL), serum calcium (mg/dL), serum phosphorus (mg/dL), serum creatinine levels (mg/dL), pre-dialysis systolic blood pressure (mmHg), pre-dialysis diastolic blood pressure (mmHg), and ultrafiltration volume (L/session), were collected monthly and averaged. We calculated $Kt/V_{urea}$ using the Daugirdas equation [7].

The patients were divided into two groups according to the prescription of spironolactone during the 6-month period of each HD quality assessment. The codes for the medications used by the patients are provided in S1 Table. Patients with no spironolactone prescription during the assessment periods were designated as the control group. Patients with one or more prescriptions of spironolactone during the assessment periods were assigned to the SPR group. The mean dose (g/day) of polystyrene sulfonate calcium (PSC) was calculated from the total dose of PSC administered over 6 months. The use of anti-hypertensive drugs, aspirin, β-blockers, renin-angiotensin blockers (RASB), and statins as concomitant medications was also evaluated. Medication use was defined as one or more prescriptions at 1 year before the evaluation in the HD Quality Assessment Program.

The presence of comorbidities at 1 year before the HD Quality Assessment Program was evaluated. Comorbidities were defined in accordance with the codes used by Quan et al. [8,9]. The Charlson Comorbidity Index (CCI), which consists of 17 comorbid conditions, was used to assess comorbidities among patients. All patients in our study were undergoing HD and considered to have renal disease. The CCI score was calculated after the patients' comorbidities were defined and identified. Furthermore, we evaluated the presence of angina, myocardial infarction, and heart failure using the following codes: I20.0, I20.1, I20.8, and I20.9 for angina; I21.0, I21.1, I21.2, I21.3, I21.4, I21.9, I22.0, I22.1, I22.8, I22.9, I23.0, I23.1, I23.2, I23.3, I23.4, I23.5, I23.6, I23.8, I24.1, and I25.2 for myocardial infarction; and I43, I50, I099, I110, I130, I132, I255, I420, I425-I429, and P290 for heart failure.

In our study, data during follow-up were collected using claims data and all patients were completely followed up to the end-point. If the patients followed up did not die by the index date (April 2022), they were defined as survivors. If the patients died before the index date, they were defined as non-survivors at the time of death. If the patients performed peritoneal dialysis or kidney transplantation before the index date, we defined the date of the first prescription for peritoneal dialysis or kidney transplantation to the end-point of follow-up and were considered survivors to the date. During the follow-up, clinical outcomes except for death were evaluated using electronic data. The codes for censoring were O7072, O7071, or O7061 for peritoneal dialysis, and R3280 for kidney transplantation. Data on patient deaths were obtained from the HIRA database.

## Statistical analyses

Data were analyzed using SAS Enterprise Guide (version 7.1; SAS Institute, Cary, NC, USA) or R (version 3.5.1; R Foundation for Statistical Computing, Vienna, Austria). Categorical variables are presented as numbers and percentages, whereas continuous variables are presented as means with standard deviations. Pearson's $\chi^2$ test or Fisher's exact test was used to analyze categorical variables. For continuous variables, means were compared using Student's t-test. Survival estimates were calculated using Kaplan–Meier curve and Cox regression analyses. *P*-values for comparison of survival curves were determined using the log-rank test. We performed multivariable analyses adjusted for covariates that showed significant differences between the two groups. Multivariable Cox regression analyses were adjusted for age, CCI

score, ultrafiltration volume, hemoglobin level, serum albumin level, serum phosphorus level, serum calcium level, systolic blood pressure, diastolic blood pressure, use of concomitant medications (anti-hypertensive drugs, β-blockers, or statins), and the presence of angina. Multivariable Cox regression analyses were performed using the enter mode. Furthermore, we performed logistic regression using the prescription of spironolactone as a dependent variable. Statistical significance was set at $P < 0.05$.

## Results

The number of patients in the control and SPR groups were 54,588 (99.4%) and 315 (0.6%), respectively. The control group had a younger mean age than the SPR group (Table 1). The SPR group had higher CCI scores and lower ultrafiltration volume, hemoglobin level, serum albumin level, serum phosphorus level, serum calcium level, serum creatinine level, systolic blood pressure, and diastolic blood pressure than the control group. The proportions of patients with use of anti-hypertensive drugs were higher in the SPR group than in the control group. The proportions of patients with angina, myocardial infarction, or congestive heart failure were higher in the SPR group than in the control group. These imply that patients with underlying heart diseases were prone to taking spironolactone. No significant difference in the mean PSC dose was observed between the two groups. Patients with β-blockers and RASB were 21370 (39.1%) and 16714 (30.6%) in the control group and 181 (57.4%) and 97 (30.8%) in the SPR group, respectively. The proportion of patients using β-blockers was higher in the SPR group than in the control group ($P < 0.001$). There was no significant difference in RASB use between the two groups ($P = 0.995$).

Multivariate logistic regression analysis showed that age, ultrafiltration volume, serum albumin, and systolic blood pressure were inversely associated with the use of spironolactone (Table 2). The CCI score, HD vintage, and the use of β-blocker use or statin were positively associated with the use of spironolactone. We included these variables as covariates in multivariate analyses for mortality.

At the follow-up end-point, the numbers of patients in the survivor, death, peritoneal dialysis, and kidney transplantation subgroups were 28,967 (53.1%), 21,291 (39.0%), 192 (0.4%), and 4,138 (7.6%) in the control group and 157 (49.8%), 142 (45.1%), 0 (0%), and 16 (5.1%) in the SPR group, respectively ($P = 0.064$). The 5-year survival rates were 69.1% and 59.1% in the control and SPR groups, respectively (Fig 1; $P < 0.001$). Cox regression analyses showed that the hazard ratio in the SPR group was 1.34 (95% confidence interval [95% CI] 1.14–1.58, $P < 0.001$) in univariate analysis and 1.13 (95% CI 0.92–1.39, $P = 0.249$) in multivariable analysis (Table 3). Univariate Cox-regression analysis showed a better patient survival rate in the control group than in the SPR group; however, multivariable analyses showed similar patient survival rates between the two groups.

In our study, 1659 patients and 3% of the control group were without spironolactone during the HD quality assessment program and with spironolactone after the program. We compared the patient survival according to patients without spironolactone during the program and follow-up, those with spironolactone during the program, and those with spironolactone during follow-up without the drug during the program. Multivariate analysis showed that the hazard ratio was 1.08 (95% CI 0.99–1.18, $P = 0.063$) for patients with spironolactone during follow-up despite no prescription during the program and 1.13 (95% CI 0.92–1.40, $P = 0.232$) for those with spironolactone during the program compared to patients without spironolactone during program and follow-up. The small proportions may not influence the overall trend.

**Table 1. Clinical characteristics of patients.**

| | Control group (*n* = 54588) | SPR group (*n* = 315) | *P*-value |
|---|---|---|---|
| Age (years) | 60.2 ± 13.0 | 62.0 ± 13.7 | 0.013 |
| Sex (male, %) | 32594 (59.7%) | 190 (60.3%) | 0.871 |
| Underlying cause of ESRD | | | 0.123 |
| Diabetes mellitus | 23990 (43.9%) | 161 (51.1%) | |
| Hypertension | 14341 (26.3%) | 68 (21.6%) | |
| Glomerulonephritis | 5771 (10.6%) | 33 (10.5%) | |
| Others | 4557 (8.3%) | 24 (7.6%) | |
| Unknown | 5929 (10.9%) | 29 (9.2%) | |
| CCI score | 7.5 ± 2.9 | 9.0 ± 3.0 | <0.001 |
| Vascular access type | | | 0.199 |
| Arteriovenous fistula | 46549 (85.3%) | 260 (82.5%) | |
| Arteriovenous graft | 8039 (14.7%) | 55 (17.5%) | |
| Kt/V$_{urea}$ | 1.5 ± 0.3 | 1.5 ± 0.3 | 0.446 |
| Ultrafiltration volume (L/session) | 2.3 ± 1.0 | 2.0 ± 0.9 | <0.001 |
| Hemoglobin (g/dL) | 10.7 ± 0.8 | 10.6 ± 0.8 | 0.043 |
| Serum albumin (g/dL) | 4.0 ± 0.3 | 3.9 ± 0.4 | <0.001 |
| Serum phosphorus (mg/dL) | 5.0 ± 1.4 | 4.6 ± 1.3 | <0.001 |
| Serum calcium (mg/dL) | 8.9 ± 0.8 | 8.8 ± 0.7 | 0.010 |
| Systolic blood pressure (mmHg) | 141 ± 16 | 137 ± 16 | <0.001 |
| Diastolic blood pressure (mmHg) | 78 ± 10 | 75 ± 10 | <0.001 |
| Serum creatinine (mg/dL) | 9.5 ± 2.7 | 7.8 ± 3.1 | <0.001 |
| PSC dose (g/day) | 3.8 ± 5.8 | 4.2 ± 7.4 | 0.315 |
| Use of antihypertensive drug | 37393 (68.5%) | 254 (80.6%) | <0.001 |
| Use of aspirin | 23460 (43.0%) | 146 (46.3%) | 0.251 |
| Use of stain | 16103 (29.5%) | 142 (45.1%) | <0.001 |
| Use of β-blockers | 21370 (39.1%) | 181 (57.4%) | <0.001 |
| Use of RASB | 16714 (30.6%) | 97 (30.8%) | 0.995 |
| The presence of angina | 23189 (42.5%) | 179 (56.3%) | <0.001 |
| The presence of MI | 4370 (8.0%) | 50 (15.9%) | <0.001 |
| The presence of HF | 22593 (41.4%) | 189 (60.0%) | <0.001 |

Data are expressed as means ± standard deviations for continuous variables and as numbers (percentages) for categorical variables. *P*-values are tested using Student's t-test for continuous variables and Pearson's $\chi^2$ test for categorical variables.

Control group, patients without spironolactone use; SPR group, patients with spironolactone use.

**Abbreviations:** CCI, Charlson comorbidity index; ESRD, end-stage renal disease; HF, heart failure; MI, myocardial infarction; PSC, polystyrene sulfonate calcium; RASB, renin-angiotensin system blockers.

## Discussion

In this study, we analyzed 54,903 patients who participated in the HD Quality Assessment Program of HIRA. The control group had a better patient survival rate than the SPR group in univariate analyses. However, a statistically significant difference was not observed after adjustment for different covariates. The mean PSC dose was similar between the two groups.

Previous studies have reported inconsistent results regarding the association between spironolactone use and clinical outcomes in patients undergoing HD. Two randomized studies in Asian populations showed better outcomes in patients with spironolactone use than in those without [10,11]. Matsumoto et al. performed a randomized trial in 309 patients with oliguria

Table 2. Logistic regression analyses for the use of spironolactone.

| | Univariate | | Multivariate | |
|---|---|---|---|---|
| | OR (95% CI) | *P* | OR (95% CI) | *P* |
| Age (years) | 1.01 (1.00–1.02) | 0.013 | 0.98 (0.97–0.99) | 0.029 |
| HD vintage (days) | 1.00 (1.00–1.01) | <0.001 | 1.00 (1.00–1.01) | 0.015 |
| CCI score | 1.18 (1.14–1.22) | <0.001 | 1.16 (1.09–1.22) | <0.001 |
| Ultrafiltration volume (L/session) | 0.74 (0.67–0.81) | <0.001 | 0.76 (0.66–0.88) | <0.001 |
| Kt/V$_{urea}$ | 0.84 (0.54–1.32) | 0.446 | 0.66 (0.36–1.22) | 0.187 |
| Hemoglobin (g/dL) | 0.86 (0.75–0.99) | 0.042 | 0.88 (0.72–1.06) | 0.177 |
| Serum albumin (g/dL) | 0.36 (0.27–0.49) | <0.001 | 0.39 (0.25–0.59) | <0.001 |
| Serum phosphorus (mg/dL) | 0.81 (0.74–0.88) | <0.001 | 0.91 (0.80–1.03) | 0.133 |
| Serum calcium (mg/dL) | 0.85 (0.75–0.96) | 0.009 | 1.02 (0.82–1.25) | 0.878 |
| Systolic blood pressure (mmHg) | 0.98 (0.97–0.99) | <0.001 | 0.98 (0.96–0.99) | <0.001 |
| Diastolic blood pressure (mmHg) | 0.97 (0.96–0.98) | <0.001 | 1.00 (0.99–1.02) | 0.663 |
| Use of antihypertensive drug | 1.91 (1.45–2.53) | <0.001 | 0.96 (0.58–1.58) | 0.874 |
| Use of β-blockers | 2.10 (1.68–2.63) | <0.001 | 2.20 (1.48–3.28) | <0.001 |
| Use of statin | 1.96 (1.57–2.45) | <0.001 | 1.88 (1.37–2.58) | <0.001 |
| The presence of angina | 1.78 (1.43–2.23) | <0.001 | 1.12 (0.80–1.55) | 0.517 |

Multivariate analysis was adjusted for age, HD vintage, ultrafiltration volume, Kt/V$_{urea}$, hemoglobin, serum albumin, serum phosphorus, serum calcium, systolic blood pressure, diastolic blood pressure, use of antihypertensive drug, β-blockers, or statin.

Control group, patients without spironolactone use; SPR group, patients with spironolactone use.

**Abbreviations:** CCI, Charlson comorbidity index; CI, confidence interval; HD, hemodialysis; OR, odds ratio.

who were undergoing maintenance HD, in which death from cardiocerebrovascular (CCV) disease or hospitalization for CCV disease was defined as primary outcome and all-cause mortality was defined as the secondary outcome [10]. They found that the primary and secondary outcomes were more favorable in patients with spironolactone use than in those without. Lin et al. analyzed 253 patients without heart failure who were undergoing dialysis and found better results in terms of the primary composite outcome, all-cause mortality, left ventricular mass index (LVMI), and left ventricular ejection fraction (LVEF) in patients with spironolactone use than in those without [11]. However, two randomized studies in Western populations did not show favorable outcomes in patients with spironolactone use [12,13]. Hammer et al. enrolled 97 patients undergoing HD and found no significant differences in LVMI, LVEF, and indicators of heart failure or physical performance between patients with and without spironolactone use [12]. They also reported a higher risk of moderate hyperkalemia in patients with spironolactone use than in those without. The SPin-D trial included 129 patients undergoing HD and compared diastolic function, LVMI, LVEF, and cardiac biomarkers between patients with and without spironolactone use [13]. However, the trial did not show the efficacy of spironolactone with respect to all of these indicators. A post-hoc study using data from the SPin-D trial also showed a higher risk of bradycardia or conduction block in patients with spironolactone use than in those without [14]. A recent meta-analysis that analyzed 15 randomized studies showed better results in terms of all-cause mortality, CCV disease events, LVMI, and LVEF in patients with spironolactone use than in those without [15]. However, the results were conflicting among studies.

Our study sheds light on some issues. First, the rate of prescription of spironolactone was very low in patients undergoing HD in South Korea. Spironolactone use might be not indicated in a large proportion of the control group, whereas the proportion of patients prescribed spironolactone was only 0.6%. Recent meta-analyses and some randomized controlled trials

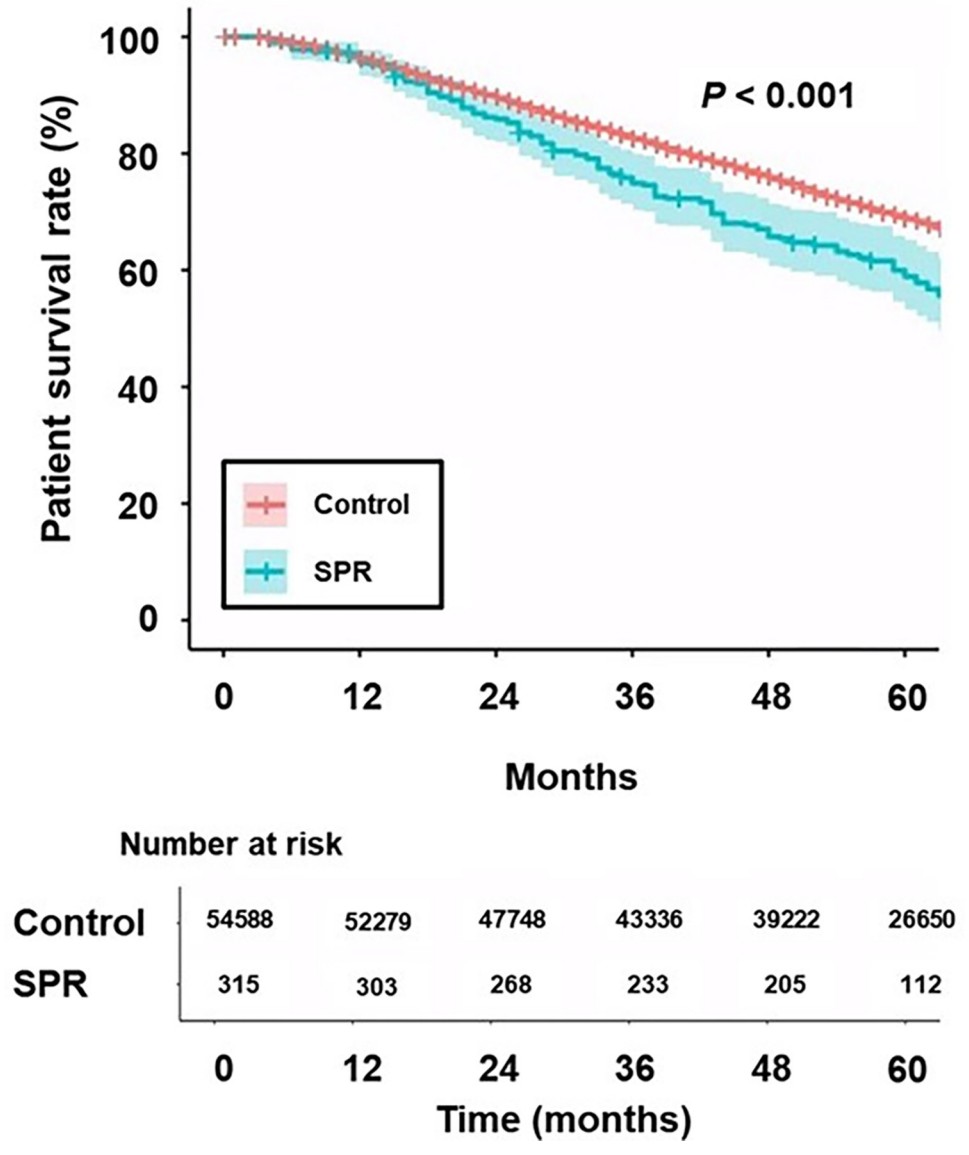

**Fig 1. Kaplan–Meier curves of patient survival in the two study groups.** SRP: Patients with prescription of spironolactone.

showed favorable results of spironolactone on clinical outcomes [10,11,15]. However, most randomized studies had small sample sizes and some studies showed a high risk of complications of gynecomastia, hyperkalemia, or arrhythmia by spironolactone [10–14]. Consequently, there was insufficient evidence regarding the clinical efficacy of spironolactone on cardiovascular outcomes or mortality beyond the hazard effect. Therefore, many clinicians may consider the use of spironolactone unnecessary. Second, the patient survival rate was better in the control group than in the SPR group in univariate analysis but not in multivariable analysis. The significant difference in patient survival in univariate analysis may be associated with the differences in baseline characteristics between the two groups. Multivariable analysis showed no association between spironolactone use and patient survival, and this result was similar to that reported in studies in Western populations. Third, the PSC dose was not significantly

**Table 3. Cox regression analyses for patient survival.**

| | Univariate | | Multivariate | |
|---|---|---|---|---|
| | **HR (95% CI)** | ***P*** | **HR (95% CI)** | ***P*** |
| Group (ref: control) | 1.34 (1.14–1.58) | <0.001 | 1.13 (0.92–1.39) | 0.249 |
| Age (increase per 1 year) | 1.06 (1.06–1.06) | <0.001 | 1.06 (1.06–1.06) | <0.001 |
| Hemodialysis vintage (increase per 1 day) | 1.00 (1.00–1.01) | <0.001 | 1.00 (1.00–1.01) | <0.001 |
| CCI score (increase per 1 score) | 1.14 (1.13–1.14) | <0.001 | 1.08 (1.08–1.09) | <0.001 |
| Ultrafiltration volume (increase per 1 kg/session) | 0.92 (0.90–0.93) | <0.001 | 1.06 (1.04–1.08) | <0.001 |
| KtV$_{urea}$ (increase per 1 unit) | 0.91 (0.87–0.97) | <0.001 | 0.68 (0.63–0.72) | <0.001 |
| Hemoglobin (increase per 1 g/dL) | 0.86 (0.85–0.88) | <0.001 | 0.92 (0.90–0.94) | <0.001 |
| Serum albumin (increase per 1 g/dL) | 0.37 (0.36–0.39) | <0.001 | 0.61 (0.58–0.64) | <0.001 |
| Serum phosphorus (increase per 1 mg/dL) | 0.85 (0.84–0.86) | <0.001 | 1.00 (0.98–1.01) | 0.759 |
| Serum calcium (increase per 1 mg/dL) | 0.94 (0.92–0.95) | <0.001 | 1.04 (1.02–1.06) | <0.001 |
| Systolic blood pressure (increase per 1 mmHg) | 1.01 (1.01–1.01) | <0.001 | 1.01 (1.01–1.01) | <0.001 |
| Diastolic blood pressure (increase per 1 mmHg) | 0.98 (0.98–0.99) | <0.001 | 1.00 (0.99–1.00) | 0.607 |
| Use of antihypertensive drug | 1.12 (1.09–1.15) | <0.001 | 0.92 (0.88–0.96) | <0.001 |
| Use of β-blockers | 1.07 (1.04–1.10) | <0.001 | 1.07 (1.03–1.12) | <0.001 |
| Use of statin | 1.10 (1.07–1.14) | <0.001 | 0.95 (0.91–0.98) | 0.004 |
| Presence of angina | 1.57 (1.53–1.62) | <0.001 | 1.10 (1.07–1.14) | <0.001 |

Multivariate analysis was adjusted for age, hemodialysis vintage, CCI score, ultrafiltration volume, Kt/V$_{urea}$, hemoglobin, serum albumin, serum phosphorus, serum calcium, systolic blood pressure, diastolic blood pressure, use of antihypertensive drug, β-blockers, or statin, the presence of angina, and was performed using enter mode.

**Abbreviations**: CCI, Charlson comorbidity index; CI, confidence interval; HR, hazard ratio.

different between the two groups, which may suggest that the incidence of adverse events associated with hyperkalemia was not largely different between the two groups.

Most studies on the association between the use of spironolactone and outcomes were performed as randomized trials and most baseline characteristics were similar between the users and non-users. To identify differences in baseline characteristics between the users and non-users, retrospective cross-sectional or observational studies would be more useful than randomized controlled trials. Lin et al. analyzed the nationwide population-based study and evaluated the use of spironolactone and clinical outcomes in patients with advanced chronic kidney disease patients. In their study, users had a higher prevalence of heart diseases, CCI scores, the use of statins or aspirin, and erythropoietin doses than non-users [16]. The results of our study showed similar trends to those of Lin's study. In our study, the use of spironolactone was inversely associated with age, ultrafiltration volume, serum albumin, systolic blood pressure, and positively associated with HD vintage, CCI scores, or the use of β-blocker or statin. Ours and Lin's studies revealed that patients with underlying heart diseases were prone to taking spironolactone and factors associated with malnutrition or volume overload were also associated with the use of spironolactone. However, due to limitations in sample size of the SPR group, caution is needed in drawing definitive information.

Our dataset did not include the prevalence of hyperkalemia or serum potassium levels. Hyperkalemia can be developed by various factors such as diet, nutritional status, or the use of many medications. In addition, the ICD-10 code may be not added despite hyperkalemia. Therefore, our study did not evaluate the effect of spironolactone on hyperkalemia. Nevertheless, we evaluated the dose of PSC, which indirectly revealed the risk of hyperkalemia or serum potassium levels. The dose of PSC was not different between the two groups. Generally, the use of spironolactone is associated with hyperkalemia. Non-difference in the use of PSC between

the two groups may be caused by some factors. First, the use of PCS did not directly reveal the hyperkalemia or serum potassium level. Serum potassium level or prevalence of hyperkalemia may be higher in the SPR group than in the control despite of same dose of PCS. In addition, the actual use of medication would not match the prescribed doses. Second, considering the risk of hyperkalemia in the SPR group, stricter diet control may be performed. Third, the SPR group had greater comorbidities, and these may be poor nutritional status. The SPR group had lower serum albumin and phosphorus levels compared to the control group.

Our study had some limitations. First, it had a retrospective design and the number of patients was largely unbalanced between the two groups. Second, the patients' comorbidities and use of medications, especially PSC, were evaluated using claims data alone. Discrepancies may be present between the registered prescriptions and the actual use of medications. In clinical practice in South Korea, PSC is not always prescribed at the time of hyperkalemia; during hyperkalemia, many patients take previously prescribed medications. Third, our study did not include data on adverse events associated with hyperkalemia, serum potassium level, etiology of spironolactone use, cause of death, or various cardiac indicators (e.g., echocardiographic parameters and cardiac biomarkers) owing to the limitations of our dataset. In our study, the outcome was all-cause mortality, and analyses using death by cardiovascular disease might show different results. Furthermore, information, such as ejection fraction, cardiac mass, or the presence of hypertrophy using echocardiography are very useful to identify the cause-relationship between the use of spironolactone and clinical outcomes beyond simple death. Moreover, left ventricular ejection fraction is very important information for classifying heart failure. Heart failure can be divided into reduced, mildly reduced, or preserved ejection fractions. The effect of spironolactone may be different according to the three groups. However, our study collected data from the HD quality assessment program and claims. HD quality assessment program only collected essential data directly regarding HD quality, such as dialysis adequacy, hemoglobin, or phosphorus levels regardless of the patient status or risk factors associated with prognosis. In addition, the death data was collected using claims data, which included data for survivor or death, but not for cause of death. These are inherent limitations of our study, being a pilot study may be valuable to provide preliminary information to design future studies.

In conclusion, this study showed no difference in survival between patients undergoing HD with and without spironolactone use. However, considering the limitations of our study, we cannot conclude that spironolactone use has no survival benefit in patients undergoing HD. Further studies with a prospective randomized design are needed to clarify this issue.

## Supporting information

**S1 Table. Medication types and health insurance review and assessment service codes.** (DOC)

## Author Contributions

**Conceptualization:** Seok Hui Kang.

**Data curation:** Seok Hui Kang, Bo Yeon Kim, Eun Jung Son, Gui Ok Kim.

**Formal analysis:** Seok Hui Kang.

**Funding acquisition:** Seok Hui Kang, Jun Young Do.

**Investigation:** Seok Hui Kang.

**Methodology:** Seok Hui Kang, Bo Yeon Kim, Eun Jung Son, Gui Ok Kim.

**Software:** Bo Yeon Kim, Eun Jung Son, Gui Ok Kim.

**Visualization:** Seok Hui Kang.

**Writing – original draft:** Seok Hui Kang, Jun Young Do.

**Writing – review & editing:** Seok Hui Kang, Jun Young Do.

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
