## [Decision Letter · Decision Letter 0]

18 Sep 2023

PONE-D-23-13967Effect of spironolactone on survival in patients undergoing maintenance hemodialysisPLOS ONE

Dear Dr. Do,

Thank you for submitting your manuscript to PLOS ONE. After careful consideration, we feel that it has merit but does not fully meet PLOS ONE’s publication criteria as it currently stands. Therefore, we invite you to submit a revised version of the manuscript that addresses the points raised during the review process.

We look forward to receiving your revised manuscript.

Kind regards,

Satoshi Higuchi

Academic Editor

PLOS ONE

Journal Requirements:

- https://www.mdpi.com/2077-0383/12/2/625

In your revision ensure you cite all your sources (including your own works), and quote or rephrase any duplicated text outside the methods section. Further consideration is dependent on these concerns being addressed.

"This work was supported by the Medical Research Center Program through the National Research Foundation (NRF) of Korea funded by the Ministry of Science, ICT, and Future Planning (2022R1A5A2018865, SHK), the Basic Science Research Program through the NRF of Korea, funded by the Ministry of Education (2022R1I1A3072966, SHK), and the NRF grant funded by the Korea government (MSIT) (2022R1F1A1076151, JYD)."

"This research was supported by a grant from the Joint Project on Quality Assessment Research, Republic of Korea. The epidemiologic data used in this study were obtained from Periodic Hemodialysis Quality Assessment by HIRA. The requirement for informed consent was waived due to the retrospective nature of the study. De-identifcation was performed, and data usage was permitted by the National Health Information Data Request Review Committee of HIRA."

"This work was supported by the Medical Research Center Program through the National Research Foundation (NRF) of Korea funded by the Ministry of Science, ICT, and Future Planning (2022R1A5A2018865, SHK), the Basic Science Research Program through the NRF of Korea, funded by the Ministry of Education (2022R1I1A3072966, SHK), and the NRF grant funded by the Korea government (MSIT) (2022R1F1A1076151, JYD). The funders had no role in the design of the study; in the collection, analyses, or interpretation of data; in the writing of the manuscript; or in the decision to publish the results."

Additional Editor Comments:

Thank you for this opportunity to review your work.

There are some comments from the Editor.

How did the authors determine each variable of CCI score? Please clarify in the Method section and quote any manuscripts referring to CCI score.

Please indicate any rationales for selection of variables in the multivariate analysis. CCI score contains renal impairment, congestive heart failure, and so on; therefore, it is inappropriate to include CCI score, serum creatinine, and congestive heart failure.

In Figure 1, please describe patient number at risk. Also, please demonstrate lost follow-up rate in the Result section.

Reviewers' comments:

Reviewer's Responses to Questions

**Comments to the Author**

1. Is the manuscript technically sound, and do the data support the conclusions?

Reviewer #1: Yes

Reviewer #2: Yes

2. Has the statistical analysis been performed appropriately and rigorously? 

Reviewer #1: Yes

Reviewer #2: No

3. Have the authors made all data underlying the findings in their manuscript fully available?

Reviewer #1: Yes

Reviewer #2: Yes

4. Is the manuscript presented in an intelligible fashion and written in standard English?

Reviewer #1: Yes

Reviewer #2: Yes

5. Review Comments to the Author

Reviewer #1: 1) Is it possible to analyze the determinants of spironolactone prescription?

2) Is it possible to mention the differences in patient background from previous studies investigating the impact of spironolactone on patients with HD?

3) In line 195, "This indicates that many clinicians hesitate about prescribing spironolactone in patients undergoing HD although it has some beneficial effects."

The benefit of spironolactone administration in patients with HD has been shown in small RCTs, but there is not sufficient evidence. Therefore, I believe that many physicians are not hesitant to prescribe spironolactone, but rather think that 'there is no need to prescribe it.

Reviewer #2: In the present work the authors present data on the impact of MRA intake on survival in patients undergoing chronic hemodialysis (HD). Even though being associated with higher mortality in a univariate analysis, the effect did not remain significant in the multivariate regression. Please find attached my comments.

1) Please provide a Table showing results of both the uni- and multivariate Cox regression model. This important to understand which parameters eliminated MRAs in the multivariate regression.

2) Can the authors provide information regarding the cause of death? Especially cardiovascular death would be of high interest.

3) Can the authors provide information on the LVEF of each group?

4) Did the patients receive other heart failure medication besides MRAs?

5) The Kaplan Meier Chart is visually not appealing. Maybe the authors could improve the design of this figure?

6) Please provide information on follow-up completeness

7) Was hyperkalemia more prevalent in the MRA cohort?

8) How did the authors deal with patients who received MRAs later during follow-up

6. PLOS authors have the option to publish the peer review history of their article (what does this mean?). If published, this will include your full peer review and any attached files.

Reviewer #1: No

Reviewer #2: No

---

## [Author Response · Author response to Decision Letter 0]

24 Nov 2023

Reviewer #1: 

1) Is it possible to analyze the determinants of spironolactone prescription?

Answer: Thank you for your comment. We have performed a logistic regression using the prescription of spironolactone as a dependent variable (Table 2). The added Table 2 is as follows:

Table 2. Logistic regression analyses for the use of spironolactone

 Univariate Multivariate

 OR (SE) P OR (SE) P

Age (years) 0.011 (0.004) 0.013 -0.022 (0.007) 0.014

HD vintage (days) -0.000 (0.000) <0.001 -0.000 (0.000) <0.001

CCI score 0.163 (0.018) <0.001 0.072 (0.029) 0.011

Ultrafiltration volume (L/session) -0.305 (0.050) <0.001 -0.175 (0.076) 0.021

Hemoglobin (g/dL) -0.146 (0.072) 0.042 -0.074 (0.090) 0.411

Serum albumin (g/dL) -1.017 (0.153) <0.001 -0.895 (0.203) <0.001

Serum phosphorus (mg/dL) -0.214 (0.045) <0.001 -0.040 (0.064) 0.534

Serum calcium (mg/dL) -0.164 (0.063) 0.009 0.043 (0.098) 0.659

Systolic blood pressure (mmHg) -0.018 (0.004) <0.001 -0.022 (0.006) <0.001

Diastolic blood pressure (mmHg) -0.030 (0.007) <0.001 -0.002 (0.009) 0.848

Serum creatinine (mg/dL) -0.251 (0.022) <0.001 -0.186 (0.035) <0.001

Use of antihypertensive drug 0.650 (0.143) <0.001 0.196 (0.235) 0.402

Use of β-blockers 0.742 (0.114) <0.001 0.737 (0.184) <0.001

Use of statin 0.674 (0.114) <0.001 0.490 (0.149) 0.001

The presence of angina 0.578 (0.114) <0.001 0.032 (0.161) 0.841

The presence of MI 0.774 (0.155) <0.001 0.574 (0.200) 0.004

The presence of HF 0.753 (0.115) <0.001 0.469 (0.163) 0.004

P-values are tested using Student’s t-test for continuous variables and Pearson’s χ2 test for categorical variables. Multivariate analysis is adjusted for age, HD vintage, ultrafiltration volume, hemoglobin, serum albumin, serum phosphorus, serum calcium, systolic blood pressure, diastolic blood pressure, serum creatinine, use of antihypertensive drug, β-blockers, or statin, the presence of angina, MI, or HF.

Control group, patients without spironolactone use; SPR group, patients with spironolactone use. 

Abbreviations: CCI, Charlson comorbidity index; HD, hemodialysis HF, heart failure; MI, myocardial infarction; OR, odds ratio; SE, standard error. 

Multivariate logistic regression analysis showed that age, HD vintage, ultrafiltration volume, serum albumin, systolic blood pressure, and serum creatinine were inversely associated with the use of spironolactone. The CCI score, the use of β-blocker or statin, or the presence of MI of HF were positively associated with the use of spironolactone. These revealed that patients with underlying heart diseases were prone to taking spironolactone. We have included these variables as covariates in multivariate analyses for mortality. We have added these comments and a Table in the Results section.

2) Is it possible to mention the differences in patient background from previous studies investigating the impact of spironolactone on patients with HD?

Answer: Thank you for your comment. Most studies for association between the use of spironolactone and outcomes were performed as randomized trials and most baseline characteristics were similar between the users and non-users. To identify differences in baseline characteristics between the users and non-users, retrospective cross-sectional or observational studies would be more useful than randomized controlled trials. Lin et al. analyzed the nationwide population-based study and evaluated the use of spironolactone and clinical outcomes in patients with advanced chronic kidney disease. In their study, users had a higher the prevalence of heart diseases, CCI scores, the use of statins or aspirin, and erythropoietin doses than non-users [1]. The results of our study showed similar trends to those of Lin’s study. In our study, the use of spironolactone was inversely associated with age, HD vintage, ultrafiltration volume, serum albumin, systolic blood pressure, and serum creatinine, and positively associated with CCI scores, the use of β-blocker or statin, or the presence of MI of HF. Ours and Lin’s studies reveal that patients with underlying heart diseases were prone to taking spironolactone and factors associated with malnutrition or volume overload were also associated with the use of spironolactone. However, due to limitations in sample size of the SPR group, caution is needed in drawing definitive information. 

Added reference

[1] Tseng WC, Liu JS, Hung SC, Kuo KL, Chen YH, Tarng DC, Hsu CC. Effect of spironolactone on the risks of mortality and hospitalization for heart failure in pre-dialysis advanced chronic kidney disease: A nationwide population-based study. Int J Cardiol. 2017 Jul 1;238:72-78. 

3) In line 195, "This indicates that many clinicians hesitate about prescribing spironolactone in patients undergoing HD although it has some beneficial effects."

The benefit of spironolactone administration in patients with HD has been shown in small RCTs, but there is not sufficient evidence. Therefore, I believe that many physicians are not hesitant to prescribe spironolactone, but rather think that 'there is no need to prescribe it.

Answer: Thank you for your comment. As the reviewer suggested, we have revised the relevant sentence as follows: 

Recent meta-analyses and some randomized controlled trials showed favorable results of spironolactone on clinical outcomes. However, most randomized studies had small sample sizes and some studies showed a high risk of complications for gynecomastia, hyperkalemia, or arrhythmia by spironolactone. Consequently, there was insufficient evidence regarding the clinical efficacy of spironolactone on cardiovascular outcomes or mortality beyond the hazard effect. Therefore, many clinicians may consider the use of spironolactone unnecessary.

Reviewer #2: In the present work the authors present data on the impact of MRA intake on survival in patients undergoing chronic hemodialysis (HD). Even though being associated with higher mortality in a univariate analysis, the effect did not remain significant in the multivariate regression. Please find attached my comments.

1) Please provide a Table showing results of both the uni- and multivariate Cox regression model. This important to understand which parameters eliminated MRAs in the multivariate regression.

Answer: Thank you for your comment. We have added Table 3 with data for covariate and univariate and multivariate analyses. Table 3 is as follows:

Table 3. Cox regression analyses for patient survival.

 Univariate Multivariate

 HR (95% CI) P HR (95% CI) P

Group (ref: control) 1.34 (1.14–1.58) <0.001 1.06 (0.87–1.30) 0.588

Age (increase per 1 year) 1.06 (1.06–1.06) <0.001 1.06 (1.06–1.06) <0.001

Hemodialysis vintage (increase per 1 day) 1.00 (1.00–1.01) <0.001 1.00 (1.00–1.01) <0.001

CCI score (increase per 1 score) 1.14 (1.13–1.14) <0.001 1.07 (1.07–1.07) <0.001

Ultrafiltration volume (increase per 1 kg/session) 0.92 (0.90–0.93) <0.001 1.09 (1.07–1.11) <0.001

KtVurea (increase per 1 unit) 0.91 (0.87–0.97) <0.001 0.64 (0.60–0.68) <0.001

Hemoglobin (increase per 1 g/dL) 0.86 (0.85–0.88) <0.001 0.92 (0.90–0.94) <0.001

Serum albumin (increase per 1 g/dL) 0.37 (0.36–0.39) <0.001 0.63 (0.60–0.66) <0.001

Serum creatinine (increase per 1 mg/dL) 0.87 (0.86–0.87) <0.001 0.95 (0.94–0.95) <0.001

Serum phosphorus (increase per 1 mg/dL) 0.85 (0.84–0.86) <0.001 1.02 (1.01–1.04) 0.001

Serum calcium (increase per 1 mg/dL) 0.94 (0.92–0.95) <0.001 1.05 (1.02–1.07) <0.001

Systolic blood pressure (increase per 1 mmHg) 1.01 (1.01–1.01) <0.001 1.01 (1.01–1.01) <0.001

Diastolic blood pressure (increase per 1 mmHg) 0.98 (0.98–0.99) <0.001 1.00 (0.99–1.00) 0.347

Use of antihypertensive drug 1.12 (1.09–1.15) <0.001 0.94 (0.90–0.98) 0.006

Use of β-blockers 1.07 (1.04–1.10) <0.001 1.07 (1.03–1.12) <0.001

Use of statin 1.10 (1.07–1.14) <0.001 0.93 (0.90–0.97) <0.001

Presence of angina 1.57 (1.53–1.62) <0.001 1.09 (1.05–1.13) <0.001

Presence of myocardial infarction 1.90 (1.82–1.98) <0.001 1.20 (1.14–1.27) <0.001

Presence of heart failure 1.43 (1.39–1.47) <0.001 1.01 (0.97–1.05) 0.576

Multivariate analysis is adjusted for age, hemodialysis vintage, CCI score, ultrafiltration volume, Kt/Vurea, hemoglobin, serum albumin, serum creatinine, serum phosphorus, serum calcium, systolic blood pressure, diastolic blood pressure, use of antihypertensive drug, β-blockers, or statin, the presence of angina of myocardial infarction or heart failure, and was performed using enter mode. 

Abbreviations: CCI, Charlson comorbidity index; CI, confidence interval; HR, hazard ratio.

2) Can the authors provide information regarding the cause of death? Especially cardiovascular death would be of high interest.

Answer: Thank you for your comment. As the reviewer pointed out, the benefits of spironolactone mainly originate from the cardiovascular system. In our study, the outcome was all-cause mortality, and analyses using death by cardiovascular disease might show different results. Furthermore, information, such as ejection fraction, cardiac mass, or the presence of hypertrophy using echocardiography are very useful to identify the cause-relationship between the use of spironolactone and clinical outcomes beyond simple death. Moreover, left ventricular ejection fraction is very important information for classifying heart failure. Heart failure can be divided into reduced, mildly reduced, or preserved ejection fractions. The effect of spironolactone may be different according to the three groups. However, our study collected from data the HD quality assessment program and claims. HD quality assessment program only collected essential data directly regarding HD quality, such as dialysis adequacy, hemoglobin, or phosphorus levels regardless of the patient status or risk factors associated with prognosis. In addition, the death data was collected using claims data, which included data for survivor or death, but not for cause of death. These are inherent limitations of our study, but our study as a pilot study may be valuable in providing preliminary information to design future studies. We have added these comments in the Discussion section.

3) Can the authors provide information on the LVEF of each group?

Answer: Thank you for your comment. We have added some comments for this issue. Detailed explanations are presented in the answer to the previous comment.

4) Did the patients receive other heart failure medication besides MRAs?

Answer: Thank you for your comment. We have added the data for β-blockers and renin-angiotensin system blockers (RASB). Patients with β-blockers and RASB were 21370 (39.1%) and 16714 (30.6%) in the control group and 181 (57.4%) and 97 (30.8%) in the SPR group, respectively. The proportion of patients using β-blockers was higher in the SPR group than in the control group (P < 0.001). There was no significant difference in RASB use between the two groups (P = 0.995). We have added these data in the Results section.

5) The Kaplan Meier Chart is visually not appealing. Maybe the authors could improve the design of this figure?

Answer: Thank you for your comment. We have revised Figure 1 to a new figure with a 95% confidence interval, P-value, and numbers at risk as follows;

Figure 1. Kaplan–Meier curves of patient survival in the two study groups. SRP: patients with a prescription of spironolactone.

6) Please provide information on follow-up completeness

Answer: Thank you for your comment. In our study, data during follow-up were collected using claims data and all patients were completely followed up to the end-point. If the patient followed up did not die by the index date (April 2022), they were defined as survivors. If the patient died before the index date, they were defined as non-survivors at the time of death. If the patients performed peritoneal dialysis or kidney transplantation before the index date, we defined the date of the first prescription for peritoneal dialysis or kidney transplantation to the end-point of follow-up and were considered survivors to the date. In addition, we have added numbers at risk in Figure 1 and the data would be useful to identify follow-up status. We have added these data in the Methods section.

7) Was hyperkalemia more prevalent in the MRA cohort?

Answer: Thank you for your comment. Our dataset did not include the prevalence of hyperkalemia or serum potassium levels. Hyperkalemia can be developed by various factors such as diet, nutritional status, or the use of many medications. In addition, the ICD-10 code was not added despite hyperkalemia. Therefore, our study did not evaluate the effect of spironolactone on hyperkalemia. Nevertheless, we evaluated the dose of polystyrene sulfonate calcium (PSC), which indirectly revealed the risk of hyperkalemia or serum potassium levels. The dose of PSC was not different between the two groups. Generally, the use of spironolactone is associated with hyperkalemia. Non-difference in the use of PSC between the two groups would be caused by some factors. First, the use of PCS did not directly reveal the hyperkalemia or serum potassium level. Serum potassium level or prevalence of hyperkalemia may be higher in the SPR group than in the control despite of same dose of PCS. The actual use of medication did not match the prescribed doses. Second, considering the risk of hyperkalemia in the SPR group, stricter diet control may be performed. Third, the SPR group had greater comorbidities, and these may be poor nutritional status. The SPR group had lower serum albumin and phosphorus levels compared to the control group. We have added these comments in the Discussion section.

8) How did the authors deal with patients who received MRAs later during follow-up

Answer: Thank you for your comment. In our study, 1659 and 3% of the control group were without spironolactone during the HD quality assessment program and with spironolactone after the program. We compared the patient survival according to patients without spironolactone during the program and follow-up, those with spironolactone during the program, and those with spironolactone during follow-up without the drug during the program. Multivariate analysis showed that hazard ratio (95% CI) was 1.08 (95% CI, 0.99-1.18; P = 0.073) for patients with spironolactone during follow-up without the drug during the program and 1.06 (95% CI, 0.86-1.31, P = 0.563) for those with spironolactone during the program compared to patients without spironolactone during program and follow-up. The small proportions of control may not influence the overall trend. We have added these comments in the Results section.

---

## [Decision Letter · Decision Letter 1]

23 Jan 2024

PONE-D-23-13967R1Effect of spironolactone on survival in patients undergoing maintenance hemodialysisPLOS ONE

Dear Dr. Do,

Thank you for submitting your manuscript to PLOS ONE. After careful consideration, we feel that it has merit but does not fully meet PLOS ONE’s publication criteria as it currently stands. Therefore, we invite you to submit a revised version of the manuscript that addresses the points raised during the review process.

We look forward to receiving your revised manuscript.

Kind regards,

Satoshi Higuchi

Academic Editor

PLOS ONE

Reviewers' comments:

Reviewer's Responses to Questions

**Comments to the Author**

1. If the authors have adequately addressed your comments raised in a previous round of review and you feel that this manuscript is now acceptable for publication, you may indicate that here to bypass the “Comments to the Author” section, enter your conflict of interest statement in the “Confidential to Editor” section, and submit your "Accept" recommendation.

Reviewer #1: (No Response)

Reviewer #2: All comments have been addressed

2. Is the manuscript technically sound, and do the data support the conclusions?

Reviewer #1: Yes

Reviewer #2: Yes

3. Has the statistical analysis been performed appropriately and rigorously? 

Reviewer #1: No

Reviewer #2: Yes

4. Have the authors made all data underlying the findings in their manuscript fully available?

Reviewer #1: Yes

Reviewer #2: Yes

5. Is the manuscript presented in an intelligible fashion and written in standard English?

Reviewer #1: Yes

Reviewer #2: Yes

6. Review Comments to the Author

Reviewer #1: If CCI (Charlson comorbidity index) is selected as a covariate when performing multivariate analysis, serum creatinine and the presence of HF or MI should not be included. This is because these items are included in the CCI. CCI should be excluded if serum creatinine and HF or MI are more preferred items.

The OR of the logistic analysis in Table 2 should not normally be negative. Is this result the slope of the regression equation?

Reviewer #2: Thank you very much for answering my and the others reviewers questions. The manuscript significantly improved.

7. PLOS authors have the option to publish the peer review history of their article (what does this mean?). If published, this will include your full peer review and any attached files.

Reviewer #1: No

Reviewer #2: No

---

## [Author Response · Author response to Decision Letter 1]

28 Feb 2024

Reviewer #1

If CCI (Charlson comorbidity index) is selected as a covariate when performing multivariate analysis, serum creatinine and the presence of HF or MI should not be included. This is because these items are included in the CCI. CCI should be excluded if serum creatinine and HF or MI are more preferred items.

Answer: Thank you for your comments. As reviewer pointed out, we have excluded serum creatinine, myocardial infarction, and congestive heart failure as covariates in the multivariate analysis. The revised results were similar with those obtained when including these variables. We have revised the results for multivariate analyses.

The OR of the logistic analysis in Table 2 should not normally be negative. Is this result the slope of the regression equation?

Answer: Thank you for your comments. In Table 2, the odd ratio (SE) are presented as regression coefficients and standard errors in logistic regression analysis. We have converted these values to more easily interpretable odds ratio with 95% confidence interval. The revised Table 2 is as follows:

Table 2. Logistic regression analyses for the use of spironolactone

 Univariate Multivariate

 OR (95% CI) P OR (95% CI) P

Age (years) 1.01 (1.00–1.02) 0.013 0.98 (0.97–0.99) 0.029

HD vintage (days) 1.00 (1.00–1.01) <0.001 1.00 (1.00–1.01) 0.015

CCI score 1.18 (1.14–1.22) <0.001 1.16 (1.09–1.22) <0.001

Ultrafiltration volume (L/session) 0.74 (0.67–0.81) <0.001 0.76 (0.66–0.88) <0.001

Kt/Vurea 0.84 (0.54–1.32) 0.446 0.66 (0.36–1.22) 0.187

Hemoglobin (g/dL) 0.86 (0.75–0.99) 0.042 0.88 (0.72–1.06) 0.177

Serum albumin (g/dL) 0.36 (0.27–0.49) <0.001 0.39 (0.25–0.59) <0.001

Serum phosphorus (mg/dL) 0.81 (0.74–0.88) <0.001 0.91 (0.80–1.03) 0.133

Serum calcium (mg/dL) 0.85 (0.75–0.96) 0.009 1.02 (0.82–1.25) 0.878

Systolic blood pressure (mmHg) 0.98 (0.97–0.99) <0.001 0.98 (0.96–0.99) <0.001

Diastolic blood pressure (mmHg) 0.97 (0.96–0.98) <0.001 1.00 (0.99–1.02) 0.663

Use of antihypertensive drug 1.91 (1.45–2.53) <0.001 0.96 (0.58–1.58) 0.874

Use of β-blockers 2.10 (1.68–2.63) <0.001 2.20 (1.48–3.28) <0.001

Use of statin 1.96 (1.57–2.45) <0.001 1.88 (1.37–2.58) <0.001

The presence of angina 1.78 (1.43–2.23) <0.001 1.12 (0.80–1.55) 0.517

Multivariate analysis was adjusted for age, HD vintage, ultrafiltration volume, Kt/Vurea, hemoglobin, serum albumin, serum phosphorus, serum calcium, systolic blood pressure, diastolic blood pressure, use of antihypertensive drug, β-blockers, or statin.

Control group, patients without spironolactone use; SPR group, patients with spironolactone use. 

Abbreviations: CCI, Charlson comorbidity index; CI, confidence interval; HD, hemodialysis; OR, odds ratio.

---

## [Decision Letter · Decision Letter 2]

18 Mar 2024

Effect of spironolactone on survival in patients undergoing maintenance hemodialysis

PONE-D-23-13967R2

Dear Dr. Do,

We’re pleased to inform you that your manuscript has been judged scientifically suitable for publication and will be formally accepted for publication once it meets all outstanding technical requirements.

Kind regards,

Satoshi Higuchi

Academic Editor

PLOS ONE

Additional Editor Comments (optional):

Reviewers' comments:

Reviewer's Responses to Questions

**Comments to the Author**

1. If the authors have adequately addressed your comments raised in a previous round of review and you feel that this manuscript is now acceptable for publication, you may indicate that here to bypass the “Comments to the Author” section, enter your conflict of interest statement in the “Confidential to Editor” section, and submit your "Accept" recommendation.

Reviewer #1: All comments have been addressed

2. Is the manuscript technically sound, and do the data support the conclusions?

Reviewer #1: (No Response)

3. Has the statistical analysis been performed appropriately and rigorously? 

Reviewer #1: (No Response)

4. Have the authors made all data underlying the findings in their manuscript fully available?

Reviewer #1: (No Response)

5. Is the manuscript presented in an intelligible fashion and written in standard English?

Reviewer #1: (No Response)

6. Review Comments to the Author

Reviewer #1: (No Response)

7. PLOS authors have the option to publish the peer review history of their article (what does this mean?). If published, this will include your full peer review and any attached files.

Reviewer #1: No

---

## [Editor Report · Acceptance letter]

21 Mar 2024

PONE-D-23-13967R2 

PLOS ONE

Dear Dr. Do, 

I'm pleased to inform you that your manuscript has been deemed suitable for publication in PLOS ONE. Congratulations! Your manuscript is now being handed over to our production team.

Kind regards, 

on behalf of

Dr. Satoshi Higuchi 

Academic Editor

PLOS ONE